# Advances in Hydrogels of Drug Delivery Systems for the Local Treatment of Brain Tumors

**DOI:** 10.3390/gels10060404

**Published:** 2024-06-17

**Authors:** Jingru Yang, Zhijie Wang, Chenyan Ma, Hongyu Tang, Haoyang Hao, Mengyao Li, Xianwei Luo, Mingxin Yang, Liang Gao, Juan Li

**Affiliations:** 1Department of Chemistry, College of Chemistry and Life Science, Beijing University of Technology, Beijing 100124, China; yangjingru@ihep.ac.cn; 2CAS Key Laboratory for Biomedical Effects of Nanomaterial and Nanosafety, Institute of High Energy Physics, Chinese Academy of Sciences (CAS), Beijing 100049, China; zjwang@ihep.ac.cn (Z.W.); macny@ihep.ac.cn (C.M.); tanghy@ihep.ac.cn (H.T.); hhyzen@sina.com (H.H.); limy@ihep.ac.cn (M.L.); luoxw@ihep.ac.cn (X.L.); yangmx@ihep.ac.cn (M.Y.)

**Keywords:** hydrogel, smart hydrogel, brain tumor, drug delivery systems, local treatment

## Abstract

The management of brain tumors presents numerous challenges, despite the employment of multimodal therapies including surgical intervention, radiotherapy, chemotherapy, and immunotherapy. Owing to the distinct location of brain tumors and the presence of the blood–brain barrier (BBB), these tumors exhibit considerable heterogeneity and invasiveness at the histological level. Recent advancements in hydrogel research for the local treatment of brain tumors have sought to overcome the primary challenge of delivering therapeutics past the BBB, thereby ensuring efficient accumulation within brain tumor tissues. This article elaborates on various hydrogel-based delivery vectors, examining their efficacy in the local treatment of brain tumors. Additionally, it reviews the fundamental principles involved in designing intelligent hydrogels that can circumvent the BBB and penetrate larger tumor areas, thereby facilitating precise, controlled drug release. Hydrogel-based drug delivery systems (DDSs) are posited to offer a groundbreaking approach to addressing the challenges and limitations inherent in traditional oncological therapies, which are significantly impeded by the unique structural and pathological characteristics of brain tumors.

## 1. Introduction

Malignant brain tumors, including prevalent types such as glioblastoma, are histologically diverse and aggressively invasive neoplasms, leading to high morbidity rates [1]. Glioblastoma is categorized as a grade IV glioma according to the World Health Organization (WHO) guidelines, with over two-thirds of primary brain tumors being aggressive in nature [2,3]. The prognosis for patients diagnosed with glioblastoma is dire, with median survival times of less than one year in half of the cases [4]. The treatment of brain tumors is particularly challenging due to their unique location and the constraints imposed by the blood–brain barrier (BBB) [5,6,7]. In recent years, there has been a growing interest in the application of hydrogels in the field of brain tumor treatment [8,9,10,11]. BCNU (carmustine) is a chemotherapy agent commonly utilized in the management of glioblastoma. However, its intravenous administration is associated with a short plasma half-life of merely 15–20 min and the potential for severe complications when plasma concentrations exceed 1400 mg [12]. To extend its therapeutic duration and mitigate adverse effects, researchers have explored alternative delivery modalities, notably employing localized therapy through the incorporation of BCNU into implantable biodegradable matrices [13]. Hydrogels, characterized by their biodegradability and excellent biocompatibility, offer a promising platform for targeted delivery directly to tumor sites, thereby significantly augmenting therapeutic efficacy [14,15,16].

Hydrogels serve as localized delivery reservoirs capable of loading drugs and precisely controlling drug release through external stimuli. For instance, Turabee and his colleagues developed a thermosensitive hydrogel incorporating nanoparticles loaded with docetaxel, thus creating a multimodal therapeutic approach [17]. This hydrogel not only delivers chemotherapeutic agents but also intelligently releases drugs to eradicate tumor cells. Additionally, hydrogels can be utilized for the delivery of immune modulators to enhance tumor immune responses. For example, Cui and colleagues designed a hydrogel containing anti-PD-1 antibodies for the local release of immune checkpoint inhibitors, thereby activating the patient’s immune system to target tumor cells [18]. This localized delivery system minimizes the systemic side effects while improving therapeutic efficacy [19,20]. Hydrogels are also employed as gene therapy carriers, facilitating the targeted delivery of gene editing tools or therapeutic genes directly to tumor cells. For instance, Chen and his colleagues utilized hydrogels to deliver the CRISPR/Cas9 system, targeting and cleaving specific oncogenes in glioblastoma [21]. This approach holds promise for directly inhibiting tumor growth through genetic modification [22,23].

Researchers continue to explore novel hydrogels with unique properties and investigate new applications. This paper provides an overview of the recent applications of hydrogels in drug delivery systems and the intelligent control of drug release, as illustrated in Figure 1 [24,25,26]. By summarizing the materials, characteristics, and application methods of hydrogels, they are categorized into injectable hydrogels, sprayable hydrogels, and implantable hydrogels [27,28,29,30]. Additionally, this article summarizes the specific applications of smart hydrogels in the treatment of brain tumors, including temperature- and pH-responsive hydrogels, photoresponsive hydrogels, and magnetic-responsive hydrogels [31,32,33,34]. Due to the rich water content and soft texture of brain tissue, hydrogels exhibit excellent biocompatibility and are more suitable for the unique environment of the brain [15]. The human body mainly consists of hydrogels and skeletal structures, which is also a crucial reason for the widespread application of hydrogels in the biomedical field. Therefore, drug delivery systems based on hydrogels hold promise for the provision of new therapeutic strategies to address the limitations of brain tumors with unique physiological and pathological structures. [35].

**Figure 1 gels-10-00404-f001:**
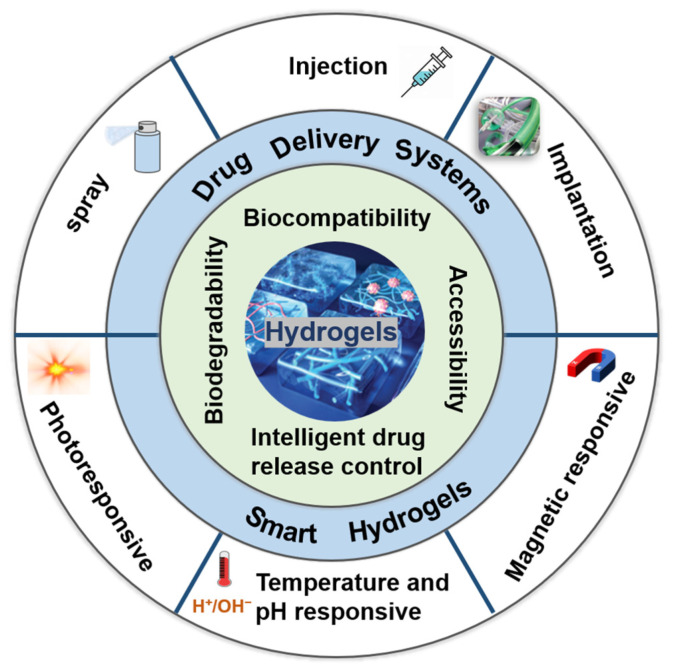
Hydrogels recently developed in drug delivery systems and intelligent drug release control for the treatment of brain tumors.

## 2. Hydrogels for the Local Treatment of Brain Tumors

Researchers have made significant progress in developing hydrogels capable of efficiently encapsulating anti-tumor drugs and releasing them in a controlled manner [36,37]. For brain tumors, hydrogels are specifically designed as precise DDSs that bypass the BBB, directly targeting the lesion and reducing systemic side effects [38,39,40,41]. The incorporation of nanoparticles within the hydrogel matrix enables targeted drug delivery. This method permits the direct injection of hydrogel into the tumor cavity or resection site, ensuring the effective delivery of therapeutic agents to tumor cells [3,42]. This section summarizes the different hydrogels used for the local treatment of cancer. The classification is based on the administration method, gelator material, and delivery characteristics (Table 1). It is evident from the table that the predominant treatment methods for brain tumors involve injections, with some emerging techniques using sprays and implantations [43,44].

### 2.1. Injectable Hydrogels

Injectable hydrogels are a specialized type of hydrogel material, typically in a liquid or semi-fluid state, which can be injected into the human tissue using a syringe or similar device, subsequently forming a gel-like state within the body. These hydrogels are commonly composed of polymer materials such as gelatin, hydroxyethyl methacrylate (HEMA), or polyethylene glycol (PEG). The notable research outlined in the literature demonstrates that Kang et al. developed an injectable, thermally responsive hydrogel nanocomposite for the treatment of glioblastoma multiforme Figure 2a [45]. Following surgical intervention, the injection of this hydrogel nanocomposite into the excised tumor site enables it to transition rapidly from a liquid to a gel state at body temperature [46]. This nanocomposite is not only responsive to thermal changes but also functions as a soft, deep intracortical reservoir for drug delivery, thereby facilitating the post-operative elimination of tumor cells [47,48].

Moreover, the surgical excision of a tumor allows for the collection of residual GBM cells by injecting biomaterials into the resection cavity [49,50]. Khan and his colleagues developed an injectable hydrogel based on polyethylene glycol and conducted studies on hydrogels with varied physical and chemical properties by manipulating parameters such as hydration level and the concentration of NaHCO_3_ in aqueous solution Figure 2b [51]. This formulation exhibits minimal and slow swelling over time, potentially reducing damage to healthy neurons post-implantation into the resection cavity. It maintains stability for up to two weeks and is both biocompatible with brain tissue and biodegradable.

In addition to acting as deep drug reservoirs, hydrogels can also stimulate anti-tumor immunity post-GBM resection, reducing recurrence [52,53]. Zhang and colleagues introduced an injectable hydrogel system containing a tumor-specific immune nanomodulator, which fosters sustained T-cell infiltration, as shown in Figure 2c [54]. When administered into the cavity of a surgically removed tumor, this hydrogel system replicates the immune ecological niche of a “hot tumor”, targeting any residual tumor cells and effectively diminishing the recurrence of GBM following surgery [55].

**Figure 2 gels-10-00404-f002:**
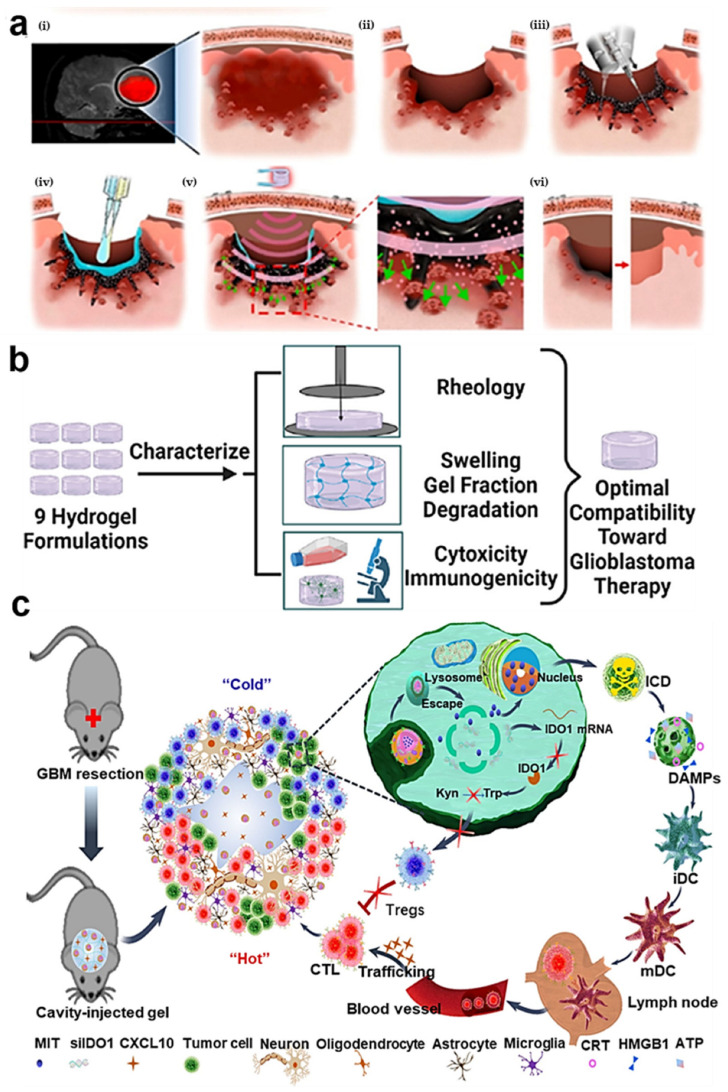
Schematic representation of injectable hydrogels for the treatment of brain tumors. (**a**) Intracortical injectable hydrogel composite nanomaterials were used to treat GBM. Adapted with permission from Kang et al. [45]. Copyright 2023 American Chemical Society. (**b**) To identify the optimal formulation for glioblastoma treatment, nine hydrogel formulations were characterized to determine the structure–property relationship between hydration/alkalinity and hydrogel properties. Adapted with permission from Khan et al. [51]. Copyright 2022 Elsevier. (**c**) A drug delivery system made of hydrogel that imitates a “hot tumor” immune niche locally. Adapted with permission from Zhang et al. [54]. Copyright 2021 Nature Research.

### 2.2. Sprayable Hydrogels

Sprayable hydrogels are hydrogels that can be applied to specific surfaces or tissues in the form of a spray. They typically consist of a gel solution or gel slurry with high water content, which can be ejected in liquid form through a sprayer or similar device and rapidly form a hydrogel network structure on the target surface or tissue [56]. Key characteristics of these hydrogels include the following: (1) Ease of application. Sprayable hydrogels can be directly sprayed onto the area requiring treatment or coating, making the application process more convenient and precise. (2) Rapid formation. Upon application to the target surface or tissue, sprayable hydrogels swiftly form a gel network structure, thereby adhering to the desired location and exerting their therapeutic effects [57]. (3) Wide applicability. Due to their ability to be sprayed onto surfaces or tissues of varying shapes and sizes, sprayable hydrogels find utility across diverse fields, including medicine, drug delivery, tissue engineering, and biomedical applications [58]. (4) Controlled release. Sprayable hydrogels can encapsulate drugs or bioactive substances, immobilizing them within the hydrogel network, and achieve the controlled release of these agents by controlling the composition and structure of the hydrogel, thereby enabling localized therapeutic effects [59]. The delivery of drugs via sprayable hydrogels represents a novel approach in the treatment of brain tumors. McCrorie and colleagues employed a spray device to effectively apply pectin and polymer nanocrystals (containing etoposide and olaparib) to sites of surgical resection Figure 3a [60]. This marks the first reported instance of transporting pectin to the brain and utilizing a spray device in neurosurgery for the local administration of drugs around the incision site. As an innovative DDS, sprayable hydrogel not only mitigates the adverse effects of surgery but also has the potential to extend patient survival [61].

### 2.3. Implantable Hydrogels

Implantable hydrogels are a type of hydrogel material that can be implanted into the human body, typically designed for releasing drugs at specific locations, supporting tissue engineering, or promoting wound healing [62]. The applications of implantable hydrogels include the following: (1) localized hyperthermia. Some implantable hydrogels contain thermosensitive components, such as magnetic nanoparticles or thermosensitive drugs. When these hydrogels are implanted into brain tissue or around tumors, external magnetic fields or other heat sources can induce the release of thermosensitive components, achieving localized hyperthermia and helping to kill tumor cells [63]. (2) Assistance in radiotherapy. Implantable hydrogels can be used as adjuncts in radiotherapy. By embedding radioactive isotopes or radiotherapeutic drugs into hydrogels and implanting them around tumors, the precision and local chemotherapy effects of radiotherapy can be enhanced. (3) Surgical assistance. Implantable hydrogels can serve as surgical adjunct materials, such as intraoperative fillers or postoperative supports. They can be implanted at surgical sites to provide support and filling, reduce surgical trauma, and promote postoperative healing [64,65]. Implantable hydrogels could enable the adjustment of drug dosage based on the tumor’s characteristics, thus optimizing treatment and minimizing side effects [20]. This approach to personalized medicine holds the potential to transform the therapy of brain tumors. Wang and his colleagues developed a novel 3D-printed hydrogel nanoplatform for intracranial implantation Figure 3b [66]. Hydrogels act as drug reservoirs, and through the modification of targeted peptides, an effective DDS can be established. They also incorporated temozolomide and erastin into gelatin methacrylamide (GelMA) to induce synergistic effects in tumor treatment. The intracranial implantation of this hydrogel liposome system can enhance the sensitivity of chemotherapy drugs and also modulate the tumor microenvironment, showing considerable promise for the treatment of brain tumors [67].

**Figure 3 gels-10-00404-f003:**
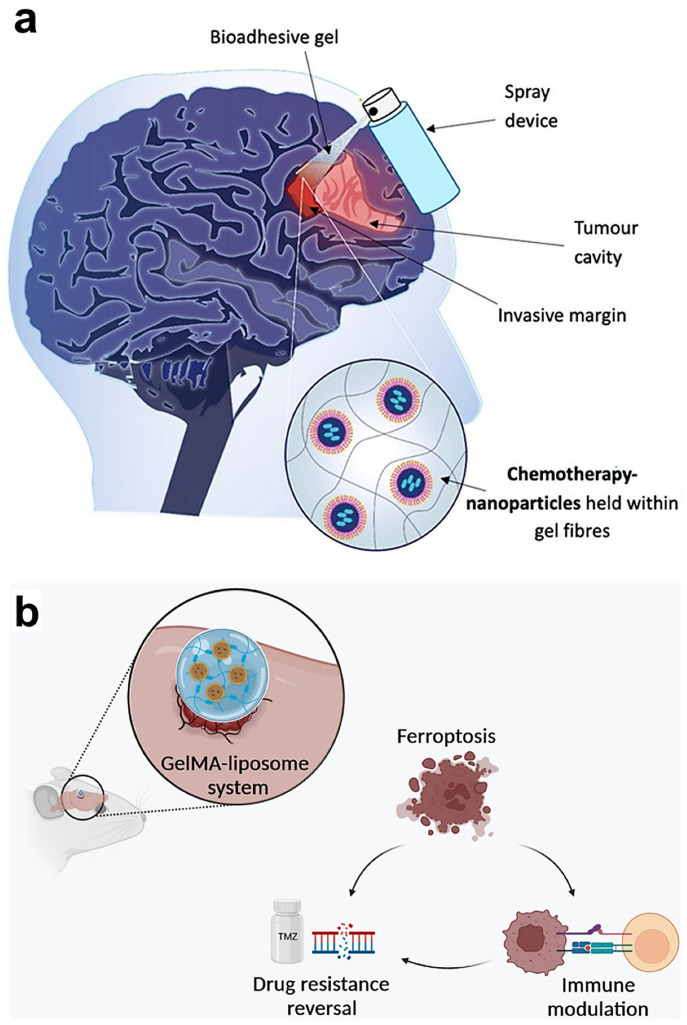
Schematic representation of hydrogels for different modes of brain tumor delivery. (**a**) Schematic illustration of a local DDS consisting of a nebulizing device, pectin, and NCPPs. Adapted with permission from McCrorie et al. [60]. Copyright 2020 Elsevier. (**b**) Schematic illustration of a GelMA–liposome system that is coated with temozolomide and erastin. Adapted with permission from Wang et al. [66]. Copyright 2023 Elsevier.

**Table 1 gels-10-00404-t001:** Methods of hydrogels for the delivery of brain tumors.

Drug Delivery Way	Hydrogel Material	Feature	Application	Ref.
Injection	Composite nanohydrogels containing drug-loaded micelles and wFIONs	Injectable heat-responsive system	Operative brain tumor therapy using injectable hydrogel nanocomposites	[45]
	Poly (ethylene glycol)-based hydrogel crosslinked by thiol-Michael addition reaction	Chemical and physical modalities were synergistically employed for therapeutic intervention	Injectable sulfhydryl Michael addition hydrogel for glioblastoma therapy	[51]
	The gelato consists of 9-fluorenylmethoxycarbonyl Phe and Phe-Phe-dihydroxyphenylalanine	Benign biodegradability and drug release properties	Tumor-killing immunity is stimulated after surgical resection of GBM to reduce its recurrence	[54]
Spray	Pectin with nanocrystals coated with polylactic acid and polyethylene glycol (NCPPs)-loaded etoposide and olaparib	Drugs are delivered using a spray device	Bioadhesive spray hydrogels containing etoposide and olaparib polymer-coated nanoparticles	[60]
Implantation	Temozolomide + Erastin@liposome-cyclic RGD + gelatin methacrylamide	The orthotopic implantation procedure elicits ferroptosis and impedes tumor recurrence	The platform of implantable hydrogels inhibits the recurrence of GBM by inducing ferroptosis	[66]

## 3. Smart Hydrogels for Local Treatment of Brain Tumors

Smart hydrogels release encapsulated bioactive substances in response to external stimuli [68,69,70,71]. Hydrogels that are responsive to temperature, pH, light, and magnetism fall into the category of smart hydrogels. These external stimuli induce changes in the properties of the hydrogels [72,73]. In the context of treating brain tumors, smart hydrogels efficiently control drug release through external stimuli, significantly reducing systemic adverse reactions due to their excellent biocompatibility [74,75,76,77].

### 3.1. Temperature and pH—Responsive Hydrogels

Temperature-responsive hydrogels demonstrate varying behaviors in response to temperature fluctuations [78,79]. Temperature-responsive polymers undergo phase transitions above ambient temperatures and dissolve at lower temperatures (lower critical solution temperature (LCST)) [80]. When the temperature rises to LCST, the polymer changes from sol to gel. However, when the temperature-responsive polymer undergoes a phase transition below ambient temperature, it dissolves at higher temperatures (upper critical solution temperature (UCST)) [81]. When the temperature is lower than UCST, the polymer turns into a gel state. In the present study, the temperature-responsive hydrogels with LCST have a good medical application prospect, mainly because the temperature-responsive hydrogels can form a gel state when they are close to human temperature. pH-responsive hydrogels are a type of smart hydrogel sensitive to changes in environmental pH. The ionizable groups within the network of pH-responsive hydrogels, such as carboxyl and amino groups, alter their dissociation states in response to environmental pH changes, thereby affecting the swelling behavior and properties of the hydrogel [82]. pH-responsive hydrogels are typically composed of polymers with acidic or basic functional groups. Common polymers include poly(acrylic acid), poly(methacrylic acid), and poly(2-vinylpyridine). pH-responsive hydrogels can be engineered to specifically deliver chemotherapeutic agents to tumor sites. Compared to normal tissues (approximately pH 7.4), the tumor microenvironment is often slightly acidic (approximately pH 6.5–6.8) [83]. Hydrogels can be designed to swell and release their drug payload in response to this acidic environment, ensuring that therapeutic agents are precisely released where needed, thus minimizing damage to healthy brain tissue [84,85,86].

Hydrogels that respond to both temperature and pH have the potential to facilitate the precise, targeted therapy of tumors under multiple stimuli, thereby proving highly effective in sustained-release applications [18]. Kang and colleagues developed a gelatin hydrogel with dual stimulus responsiveness, grafted with oligomeric sulfadiazine (OSM) and combined with paclitaxel (PTX) to inhibit GBM progression Figure 4a [87]. This gelatin-OSM complex transitions from a fluid to a gel state dependent on temperature and pH, maintaining its gel state for approximately ten days. The dual-responsive hydrogel thus provides sustained drug release within the tumor environment, effectively impeding GBM progression.

### 3.2. Photoresponsive Hydrogels

In the near infrared–ultraviolet/visible (NIR-UV/VIS) spectrum, water is nearly transparent [88,89]. Photoresponsive hydrogels alter their morphology in response to photo irradiation through the absorption or release of water [90,91,92]. Consequently, hydrogels are well-suited as photoresponsive biomaterials. The chemical and physical versatility of hydrogels, combined with their photoresponsiveness, makes them ideal for a range of applications, spanning from biomaterials to biomedicine [93]. Zhao and colleagues explored a local DDS based on photopolymerizable hydrogels for postoperative GBM treatment Figure 4b [94]. Upon photo irradiation, the hydrogel not only forms rapidly but also exhibits low swelling, thereby preventing an increase in intracranial pressure [95,96]. To enhance the therapeutic efficacy against GBM, they incorporated paclitaxel (PTX) and temozolomide (TMZ) into the hydrogels to create a combined DDS [97,98]. In a U87MG orthotopic transplantation tumor model, the hydrogel proved suitable for implantation post-tumor resection, demonstrating the excellent sustained-release capabilities of the drug [99,100,101]. There has been a marked increase in the development of photoresponsive hydrogels in recent years [102]. These hydrogels possess physical and chemical characteristics akin to the flexible materials found in living systems [103]. One of the primary advantages of photoresponsive hydrogels is their cost-effectiveness, coupled with their ability to achieve non-contact and spatiotemporal control. These properties render them excellent candidates for applications in biomaterials and biomedicine [104].

### 3.3. Magnetic-Responsive Hydrogels

Magnetic-responsive hydrogels are typically fabricated by incorporating micron or nanometer-sized magnetic particles, such as Fe_2_O_3_ and Fe_3_O_4_ [105,106]. When subjected to an external magnetic field, the hydrogel’s magnetic nanoparticles can be released with precision [107]. Magnetic fields offer advantages over other stimuli as they are contactless and relatively straightforward to manipulate, making them particularly suitable for biomedical applications [108]. Kang and colleagues have recently reported the use of magnetically responsive hydrogels in the treatment of brain tumors Figure 4c [45]. Following surgical intervention, the hydrogel, injected into the site of the excised brain tumor, rapidly transitions to a gel state at body temperature. Under alternating magnetic fields, the hydrogel, mixed with water-dispersible ferrimagnetic iron oxide nanocubes (wFIONs), generates heat, accelerating the micelle process. Consequently, the release and diffusion of the drug can penetrate a centimeter deep, facilitating precise drug delivery for brain tumor treatment. Magnetic-responsive hydrogels hold significant potential for targeted tumor therapy, with diverse applications including magnetically controlled drug release, magnetic hyperthermia, and magnetic targeting [109].

**Figure 4 gels-10-00404-f004:**
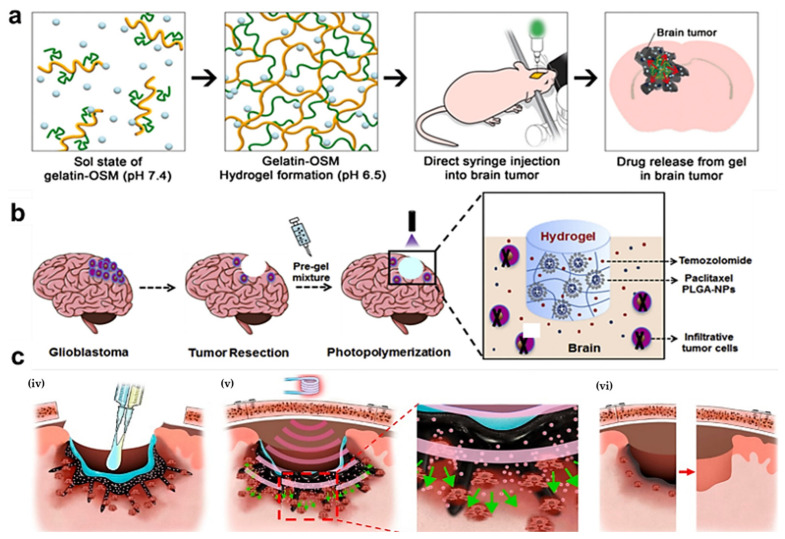
Schematic representation of smart hydrogels for the treatment of brain tumors. (**a**) In situ injection of gelatin–OSM hydrogel to inhibit tumor recurrence. Adapted with permission from Kang et al. [87]. Copyright 2021 Elsevier. (**b**) Codelivery of PTX and TMZ through a photoresponsive hydrogel for the postresection therapy of GBM. Adapted with permission from Zhao et al. [94]. Copyright 2019 Elsevier. (**c**) The magnetic-responsive hydrogel precisely delivers wFIONs deep into the brain tumor. Adapted with permission from Kang et al. [45]. Copyright 2023 American Chemical Society.

## 4. Advantages of Hydrogels in the Treatment of Brain Tumors

Despite significant advances in biomedicine, malignant brain tumors remain a formidable challenge, with a cure remaining elusive. The BBB significantly impedes therapeutic efficacy, primarily by restricting the entry of large molecules and over 90% of small molecular drugs into the brain [110,111,112,113]. Additionally, the invasive nature of brain tumors further complicates treatment, with tumor cells infiltrating surrounding healthy brain parenchyma and developing mechanisms of multidrug resistance. These factors collectively pose substantial challenges in brain tumor therapy [114].

Hydrogels exhibit remarkable properties stemming from their crosslinked polymer networks, which enable them to retain substantial amounts of water within their structure [115,116]. As depicted in Figure 5, hydrogels can be classified into physical, chemical, and dual network hydrogels based on their crosslinking mechanisms [117]. Table 2 summarizes the advantages and disadvantages of the different types of hydrogels. The crosslinking in physical hydrogels primarily occurs through physical interactions such as hydrophobic association, chain aggregation, crystallization, polymer chain complexation, and hydrogen bonding [118]. The shortcomings of physical hydrogels manifest in several ways: (1) Low mechanical strength. Due to their weaker crosslinking interactions, these hydrogels have reduced mechanical strength and stability. (2) Lower stability. Physical crosslinking points, formed by weaker interactions like hydrogen bonds, may lead to long-term stability issues such as point fracture. (3) Limited control. Achieving precise control over crosslinking density and structure is challenging, hindering the finetuning of the hydrogel’s properties [119]. Despite these shortcomings, physical hydrogels offer the following notable advantages: (1) Mild conditions and a simple preparation: These hydrogels can form at room temperature, with synthesis easily adjusted by alterations in the solution temperature or pH, aiding cost-effective large-scale production. (2) Reversibility. Their reversible nature allows for swelling or shrinking in response to external changes, facilitating reuse. (3) Tunability. Various properties can be achieved by selecting different physical crosslinking mechanisms and adjusting conditions [120].

**Figure 5 gels-10-00404-f005:**
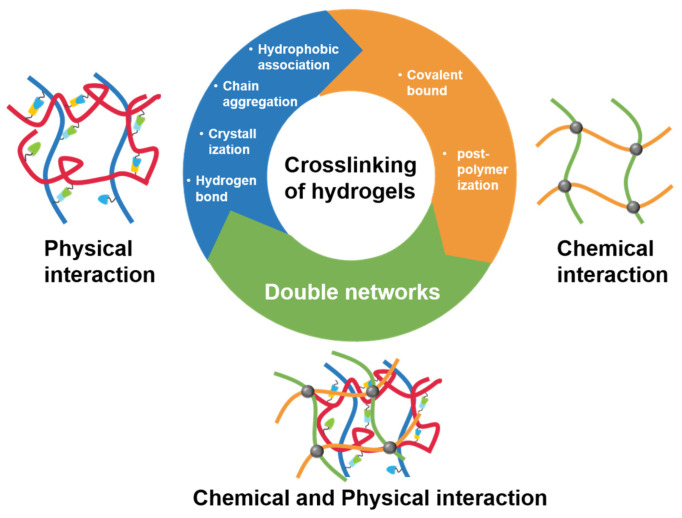
Hydrogels are prepared by physical crosslinking, chemical crosslinking, and dual networks.

Chemical hydrogels, on the other hand, are synthesized through covalent crosslinking or post-polymerization. Chemical hydrogels may present the following drawbacks: (1) Potential toxicity. Chemical crosslinking agents, such as formaldehyde and glutaraldehyde, can be toxic. In biomedical applications, it is imperative to ensure that residual levels of chemical crosslinking agents are minimized to non-toxic levels [121]. (2) Complex preparation. Chemical crosslinking processes typically require precise control over the reaction temperature and time, and may involve multiple steps of chemical processing. (3) Poor reversibility. Chemical hydrogels form crosslinking points through covalent bonds, making it challenging to reverse their structure and morphology again. In addition, the advantages of chemical hydrogels are as follows: (1) High mechanical strength. Chemical hydrogels exhibit an elevated mechanical strength and elastic modulus due to their stable covalent bond formation [122]. (2) Stability. They maintain their structural integrity and performance across diverse environmental conditions, making them suitable for long-term applications like tissue engineering scaffolds and sustained drug release systems. (3) Tunability. With varied crosslinking agents and conditions, the crosslinking density and pore structure of chemical hydrogels can be precisely controlled, enabling their properties to be tailored for specific applications [123,124,125]. Dual-network hydrogels integrate the strengths of both physical hydrogels and chemical hydrogels via the following characteristics: (1) High strength and toughness. They leverage a rigid, brittle first network for strength and a soft, stretchable second network for toughness, resulting in superior performance [126]. (2) High elasticity. These hydrogels maintain elasticity even under significant deformation, making them ideal for soft tissue engineering and biomimetic materials [127]. (3) Versatility. By adjusting the composition and crosslinking methods of both networks, multifunctional properties such as self-healing, conductivity, and responsiveness can be achieved. [128,129,130]. Certainly, dual-network hydrogels also exhibit certain disadvantages. For instance, there are limitations in material selection. Achieving synergy between the two networks necessitates stringent requirements for material compatibility and crosslinking conditions. By summarizing the unique advantages and drawbacks of each type of hydrogel, researchers can select suitable hydrogel materials according to their specific needs. This enables the more effective utilization of hydrogels in biomedical applications [131].

As exemplary biocompatible materials, hydrogels not only emulate the extracellular matrix in the brain but also establish ideal DDSs for the local treatment of brain tumors [132,133]. Hydrogels offer the following advantages as DDSs: (1) Biocompatibility. Hydrogels closely resemble human tissues in their properties. When drugs are encapsulated within hydrogels, they not only avert the rapid degradation of chemotherapy drugs in the body but also shield the brain from drug-related toxicity [134,135]. (2) Biodegradability. Ideally, the interaction between the host tissue and the hydrogel should orchestrate and finetune the degradation process, leading to its eventual disappearance [136]. (3) Intelligent drug release control. Smart hydrogels are particularly effective for targeting tumors due to their capacity to respond to various external stimuli and precisely regulate drug release [137,138,139]. (4) Accessibility. Hydrogels can be easily synthesized and mass-produced through chemical methods with a low ecological impact [140,141,142]. Currently, researchers in the field of hydrogels are developing multiple types and modes of hydrogel drug delivery systems, aiming to significantly contribute to the clinical treatment of brain tumors [143,144,145,146].

**Table 2 gels-10-00404-t002:** Properties of hydrogels with different mechanisms.

Various Types of Hydrogels	Crosslinking Mechanisms	Advantages	Disadvantages
Physical hydrogels	Hydrophobic association Chain aggregation Crystallization Polymer chain complexation Hydrogen bonding	Mild conditions Reversibility Tunability	Low mechanical strength Lower stability Limited control
Chemical hydrogels	Covalent crosslinking Post-polymerization.	High mechanical strength Stability Tunability	Potential toxicity Complex preparation Poor reversibility
Dual-network hydrogels	Physical and chemical interaction	High strength and toughness High elasticity Versatility	Limitations in material selection Elaborate synthesis steps

## 5. Conclusions and Prospectives

Due to the complexity of the brain anatomy and the high sensitivity of the nervous system, achieving effective treatment for tumors remains challenging. Some contributing factors include the following: (1) Risk of local tissue damage. There is a potential risk of local tissue damage during the injection of hydrogels, especially during the use of injection needles and surgical procedures. (2) Uneven drug distribution. Due to the nature of local treatment, drug concentrations within tumor tissues may be uneven, leading to suboptimal treatment effects in certain areas. (3) Long-term safety concerns. The long-term safety of local hydrogel therapy for brain tumors remains incompletely understood, particularly regarding the prolonged release of drugs and their impact on the brain tissue and the surrounding environment, necessitating long-term observation and research.

This paper summarizes four key aspects: (1) Challenges in the treatment of brain tumors. (2) The use of drug delivery systems using different hydrogels, including injectable hydrogels, spray hydrogels, and implantable hydrogels, detailing their material composition, characteristics, and specific applications. (3) The widespread application of various types of smart hydrogels, such as temperature- and pH-responsive, photoresponsive, and magnetic-responsive hydrogels, in the treatment of brain tumors. (4) The advantages and disadvantages of hydrogels formed by different crosslinking methods. Through this summary, it is evident that while hydrogels face challenges in the clinical treatment of brain tumors, their unique properties hold promise for postoperative treatment following GBM resection. Additionally, smart hydrogels have significant potential in enhancing treatment efficacy and reducing side effects. The development of more sophisticated hydrogels and their optimization for various medical applications represents an actively explored field, indicating a promising future for personalized medicine.

## Data Availability

The original data presented in the study are openly available in Figure 2a at Ref. [45], Figure 2b at Ref. [51], Figure 2c at Ref. [54], Figure 3a at Ref. [60], Figure 3b at Ref. [66], Figure 4a at Ref. [87], Figure 4b at Ref. [94] and Figure 4c at Ref. [45].

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
