# Peer review of "Advances in Hydrogels of Drug Delivery Systems for the Local Treatment of Brain Tumors"

_gels, 2024, doi:10.3390/gels10060404_

Round 1

Reviewer 1 Report

Comments and Suggestions for Authors

1. Introduction:

The author should provide recent statistics on brain tumors and the treatments that are currently used for the management of brain tumors along with their limitations. Further, the advantage of hydrogels for the local treatment of brain tumors should be explained. Furthermore, the author should briefly discuss the various types of hydrogels used for the management of brain cancer along with their unique properties. 

"Recent research has identified hydrogels as promising therapeutic 33 strategies and drug delivery systems (DDSs) for the local treatment of brain tumors" This line seems vague. As per my knowledge, hydrogels are the type of DDS. Please modify this. 

2. Different DDSs for local treatment of brain tumors: this heading should be Hydrogels for the local treatment of brain tumors.

"This paper summarizes the use of hydrogels in the treatment of brain tumors, categorizing them by delivery method, gelator material, and delivery characteristics, as detailed in Table 1": This line should be restructured. My suggestion is: This section summarizes different hydrogels for the local treatment of cancer. The classification is based on the administration method, gelator material, and delivery characteristics (Table 1).

Further, all the subsections like injectable, spray, and implantable hydrogels should comprise the unique characteristics of each hydrogel along with some recent examples. Moreover, the research data should be presented in tabular form comprising the columns composition of hydrogel, method of preparation, advantage or unique properties, result outcome, and references. 

3. Smart hydrogels for local treatment of brain tumors: Here the author should explain about the smart hydrogels and their classification. Moreover, the research data should be presented in tabular form comprising the columns composition of hydrogel, method of preparation, advantage or unique properties, result outcome, and references. 

4. The author should modify all the captions used for figures and tables. Moreover, the permissions (if any) should also be placed in the figures. 

5. The author should add challenges sections also. 

Comments on the Quality of English Language

The English is poor. The author should thoroughly revise the manuscript and especially focus on the sentence structure. 

Reviewer 2 Report

Comments and Suggestions for Authors

The review paper entitled "Advances in Hydrogels of Drug Delivery Systems for Local Treatment of Brain Tumors", in my opinion, is not sufficiently sound for publication. I think the authors should address the following major issues in the manuscript before it can be further recommended for publication.

1.         English is modest. Therefore, the authors need to improve their writing style. In addition, the whole manuscript needs to be checked by native English speakers.

2.         What is the novelty and difference of your review compared to other published manuscripts? The authors of this study need to present more highlights.

3.         Although the authors reference many relevant papers in the field, I think reference 8 is not related to the drug delivery systems (DDSs) for local treatment, especially in brain tumors. Authors should justify this issue.

4.         The introduction should be improved and needs more literature review. Expand on the potential opportunities that the hydrogels could open up for local treatment of brain tumors applications.

5.         The sentences on Page 1, Line 33, and Line 37 are the same. Provide different and more information in the introduction section.

6.         Each part (section 2.1) should start with an introduction-like statement and then discuss clearly other studies and works.

7.                  Provide a clear summary of the challenges faced in brain tumor treatment and introduce the proposed hydrogels.

8.                  Insert references of each study in Table 1 and move the Table to the end of section 2.

9.                  Figures should be replaced by higher-resolution ones.

10.              I think the manuscript should pay more attention to focus on the in vivo studies and applications.

11.              More discussion should be given for the favorable design of hydrogels in the perspective part.

12.              It would be helpful to outline the potential advantages and disadvantages of each type of technology and methods for hydrogel preparations.

13.              It should be pointed out that the evidence provided is not enough to support the conclusion in this manuscript. To make the conclusion section more informative, it must be modified to consider future perspectives and discuss the evidence presented in the manuscript.

Comments on the Quality of English Language

English is modest. Therefore, the authors need to improve their writing style. In addition, the whole manuscript needs to be checked by native English speakers.

Reviewer 3 Report

Comments and Suggestions for Authors

The present manuscript entitled “Advances in Hydrogels of Drug Delivery Systems for Local Treatment of Brain Tumors” provides thorough insight into the use of hydrogels as an emerging candidate for targeted drug delivery applications in the treatment of brain tumors. The manuscript covers various aspects relevant to the medicinal and pharmaceutical industries. The author's discussion and extensive literature survey are well-explained and comprehensive. Therefore, I believe that this article can be accepted after addressing the following detailed revisions:

1.     The English language used throughout the manuscript appears overly elaborate, which may hinder readability and comprehension. It would be beneficial if the authors could simplify the language to some extent. Simplifying the text will make the content more accessible and easier to understand for a broader audience, while still conveying the necessary scientific information accurately. Please focus on clear and direct expressions to enhance the manuscript's readability.

2.     The phrase “this paper” is used excessively in the last paragraph of the introduction, leading to repetition. To improve the readability and flow of this section, it is suggested that the authors rephrase these sentences in a more creative and varied manner. Consider summarizing the aims and scope of the study in a concise yet innovative way to avoid redundancy.

3.     In Fig. 1, the symbols H+ and OH- appear to represent temperature and pH, respectively, and serve as a thermostat for both. To improve clarity and ensure direct understanding, it is recommended to rearrange these symbolic representations to more appropriate locations within the figure. This adjustment will help readers immediately grasp the intended meanings of these symbols and the relationships they represent within the context of the figure.

4.     In Table 1, where different techniques and their applications in brain tumor treatment are presented, it is advisable to add an additional column for citing appropriate references. Including citations for each technique and application will allow readers to refer to the original sources for more detailed information. This will enhance the table's utility as a cross-reference tool, enabling viewers to easily follow up on the cited articles and explore the reported applications in greater depth.

5.     In line no. 129, where the manuscript discusses various types of stimuli important for hydrogels in drug delivery applications, please include pH as an additional stimulus. pH is one of the most important stimuli for targeted drug delivery, especially in environments with varying acidity, such as different bodily tissues or diseased areas. Adding pH to the list will provide a more comprehensive overview of the stimuli-responsive behaviors of hydrogels.

6.     In lines 136-137, where the authors discuss temperature-responsive behavior of polymeric solutions, please introduce the terms Lower Critical Solution Temperature as LCST for below the critical temperature and Upper Critical Solution Temperature as UCST for above the critical temperature. These terms are commonly used in the field of polymer science and will help to accurately convey the temperature-dependent phase behavior of polymeric solutions to the readers.

7.     To avoid confusion, please mention somewhere in the manuscript that "photo-responsive" and "light-responsive" are used interchangeably. The authors can standardize the terminology by writing "light/photo-responsive" wherever applicable.

8.     The paragraph in lines 156 to 161 should include appropriate references to support the claims and information presented.

9.     While the conclusion is well-written, it lacks a detailed discussion on future perspectives. Please elaborate on how hydrogel-based technologies are expected to evolve for use in brain tumor treatments and targeted drug delivery applications. Discuss potential advancements, emerging trends, and how ongoing research might address current limitations. Highlighting these future directions will provide readers with a comprehensive understanding of the potential impact and ongoing development in this field.

Comments on the Quality of English Language

The English language used throughout the manuscript appears overly elaborate, which may hinder readability and comprehension. It would be beneficial if the authors could simplify the language to some extent. Simplifying the text will make the content more accessible and easier to understand for a broader audience, while still conveying the necessary scientific information accurately. Please focus on clear and direct expressions to enhance the manuscript's readability.

Round 2

Reviewer 1 Report

Comments and Suggestions for Authors

Now The manuscript is ok.

Reviewer 2 Report

Comments and Suggestions for Authors

The manuscript is acceptable in this format.